# Prevalence, causes, and factors associated with obstructed labour among mothers who gave birth at public health facilities in Mojo Town, Central Ethiopia, 2019: A cross-sectional study

**Tarekegn Girma[1], Wubishet Gezimu[ID][2]\*, Ababo Demeke[3]**

**1** Mojo Primary Hospital, Mojo, Central Ethiopia, **2** Department of Nursing, College of Health Sciences, Mettu University, Mettu, Ethiopia, **3** Department of Nursing, College of Health and Medical Sciences, Dilla University, Dilla, Ethiopia

\* wubishet151@gmail.com

**Data Availability Statement:** All relevant data are within the paper and its Supporting information files.

## Abstract

### Background

Obstructed labour is a type of abnormal labour that is one of the causes of obstetric complications such as maternal and fetal mortality and morbidity. Early detection is the key to reducing complications.

### Objective

This study aimed to assess the prevalence, causes, and factors associated with obstructed labor among mothers who gave birth at public health facilities in Mojo Town, Central Ethiopia.

### Methods

An institution-based cross-sectional study was conducted from November 10 to December 30, 2019 among 318 women who gave birth at public health facilities in Mojo Town. Face-to-face interviews and participants' medical record reviews were utilized to gather data. The collected data were checked, coded, and entered into EpiData version 3.1 and then exported to SPSS version 23 for analysis. A binary logistic regression model was used to test the association between the dependent and independent variables. In bivariate analysis, all variables with a p-value less than 0.25 were included in multivariate analysis. Finally, a significant statistical association was declared at a p-value less than 0.05.

### Results

The prevalence of obstructed labour in this study was 51 (16%), and cephalo-pelvic disproportion (66%), mal-presentation (22%), and mal-position (12%) were reported as causes of obstructed labour. Primgravidity (AOR = 7.74: 95%CI = 2.13, 18.2) and a one-time antenatal

**Funding:** The author(s) received no specific funding for this work.

**Competing interests:** The authors have declared that no competing interests exist.

**Abbreviations: ANC**, **A**ntenatal **C**are; **CPD**, **C**ephalo-**p**elvic **D**isproportion; **MM**, **M**aternal **M**ortality; **OL**, **O**bstructed **L**abour.

care follow-up (AOR = 9.50: 95%CI: 1.91, 33.07) were found to be associated factors with obstructed labour, while labour duration of 12–24 hours (AOR = 0.20: 95%CI = 0.17, 0.87) was identified as a factor decreasing the risk of obstructed labour.

## Conclusion

The prevalence of obstructed labour in this study was higher than in the majority of previous similar local and global studies. In this study setting, cephalo-pelvic disproportion, mal-presentation, and mal-position were found to be the causes of obstetric labour. Additionally, factors such as gravidity, frequency of antenatal follow-up, and duration of labour were significantly associated with obstructed labour. Therefore, the concerned entities need to work to curb young age pregnancy as well as to strengthen counselling mothers on the importance of subsequent antenatal-follows in the prevention of obstructed labour.

## Introduction

Obstructed labour (OL) is defined as labour that does not advance despite adequate uterine contractions because fetal size is out of proportion to the mother's birth canal [1–3]. The presence of slow cervical dilatation, sluggish or no descent, and the development of pathological rings in the lower uterine segment all point to the diagnosis of OL [3].

OL has negative effects on both the mother and her fetus if it is neglected, not properly recognized, or not treated. It is one of the obstetrical tragedies that result in labour complications like infection, damage to nearby tissues, uterine rupture, and the mother's death from hemorrhagic shock. Additionally, it results in stillbirth and infant hypoxia [4]. Furthermore, OL results in obstetric fistula, the most common obstetric morbidity, as a long-term consequence [5–8].

The burden of OL is outmoded in economically advanced countries. It is, however, still high in countries with limited access to obstetric care [1,9]. In resource-limited countries, OL accounted for 22% and 9% of pregnancy complications and maternal mortality (MM), respectively. Sub-Saharan Africa's region, including Ethiopia, was responsible for nearly one-quarter (24%) of MM [10]. In Ethiopia alone, it accounted for 17.3% of MM and 39.7% of stillbirths [4,11]. The incidence and prevalence of OL vary geographically in Ethiopia. For instance, it was 3.3% and 12.2% in the northern and southwestern parts of the country, respectively [12,13].

The causes of OL are mechanical factors that disproportionately affect the passenger and the pathway (birth canal). These factors include cephalo-pelvic disproportion (CPD), mal-presentation, and malposition, which are the most common causes of OL [12,14–19]. The rare causes of OL include locked twins, fetal anomalies, and maternal soft tissue tumors such as fibroids [3,17,20].

According to the scientific facts, socio-demographic features, obstetric characteristics, and healthcare facility-related factors were shown to be associated with OL. The age of the mother, place of residence, and level of education were socio-demographic features associated with OL [1,4,21–24]. Based on obstetrical factors, gravidity, frequency of antenatal visits, and birth to big baby were all associated with OL [1,22–26]. Distance from the health facility, use of partographs, and duration of labor were all health facility-related factors associated with OL [1,11,22,24,27].

Fortunately, OL is a preventable obstetric hazard. Strategies that include increasing maternal knowledge of obstetric danger signs, birth preparedness, and skilled delivery [26,28] and adequate childhood and adulthood nutritional intake are important to reduce obstructed labour [1]. Furthermore, pelvic assessment, risk identification, early diagnosis of mal-presentation/malposition, measuring the descent of the fetal presenting part, labour follow-up with partograph, and vacuum extraction are basic health care practitioner skills required for OL management and prevention [29,30].

Despite many strategies, such as the establishment of maternal waiting homes and improved access to comprehensive emergency obstetric care (CEmOC), the burden of OL is still high in Ethiopia [27]. There has been a paucity of data on the OL in our study area, especially in central Ethiopia. Therefore, the aims of this study were to assess the prevalence of OL, to identify its causes, as well as to determine factors associated with OL among mothers who gave birth in Mojo Town.

## Methods and materials

### Study design, setting and period

An institution-based cross-sectional study was conducted in Mojo town from November 10 to December 30, 2019. Mojo town is located in the East Shoa Zone of Oromia Regional State, Central Ethiopia, which is 77 km away from Addis Ababa. In the town, there was one public hospital, one private hospital, and three health centres. Actually, this study was conducted at Mojo hospital, which is one of the public facilities in Mojo town. Mojo Hospital has been offering a variety of services to clients referred by local health canters. The estimated catchment population of the hospital was around 357,095 thousand clients. Of these, males account for 181698 and 17539 females. A total of 155 medical and supportive staff were given service at the hospital. Concerning the obstetrics and gynaecology services, the hospital had 12 midwives, 3 Integrated Emergency Surgery and Obstetrics (IESO), and it had 9 beds in the gynaecology ward and 11 beds in both labour and delivery rooms. The estimated annual average number of delivery cases in Mojo hospital was 1838.

### Population and eligibility criteria

In this study, all women who gave birth at public health facilities in Mojo Town were considered as the source population, and women who were systematically selected for the study were the study population. However, women who were severely sick during data collection and those who were referred to other facilities were excluded from this study. Likewise, women who delivered by elective cesarean section were excluded.

### Sample size calculation

The sample size for the first objective was determined using the single population proportion formula and the assumptions used were: a 95% confidence interval (CI); 4% margin of error; and a population proportion of 34.3%, which was taken from a study conducted in Western Harerghe zone public hospitals [27].

$$n = \frac{Z^2 1 - ^{\alpha}/_2\, p(1-p)}{d^2}; n = \frac{(1.96)^2 (0.343)(1 - 0.343)}{0.04^2} = 344$$

We have used a correction formula of nf = 344/(1 + 344/1836) = 289, and a response rate of 10%. So, the final calculated sample size was n = 318. The sample size for the second objective was calculated by using epi info version 7.2. Assumptions such as 95%CI, 80% power, 0.69

odds ratio, and the age of the mother being less than 19 were associated with obstructed labor (taken from a study conducted in Halaba Kulito primary hospital [21]. Hence, the total sample size for the second objective with a 10% non-response rate was n = 299. The total number of participants in the study was n = 318, which exceeded the n = 299 sample size for the second aim.

## Sampling technique

Mojo Town has three public health facilities (a hospital and two health centers), and we randomly selected Mojo Hospital (35% of the facilities) in order to ensure the sample is representative and economic. Then a consecutive sampling technique was used to select the study participants in the postpartum unit. Since the labour cases came from different socio-demographic backgrounds (with unique characteristics) in the town, we assumed that the study unit was representative of the source population.

## Study variables

Fig 1 describes the outcome and explanatory variables of this study. Obstructed labor and causes of obstructed labour were the outcome variables, while socio-demographic features, obstetric characteristics, labour outcomes, and healthcare facility-related factors were considered as explanatory variables.

## Operational definitions

**Obstructed labor**: is referred as failure of descent of the fetus in the birth canal for mechanical reasons in spite of good uterine contraction [1,2,31,32].

**Malposition**: any position of the vertex other than occipito-anterior (occipito-posterior and occipito-transverse) [31,32].

**Cephalo-pelvic disproportion (CPD)**: Refers to a mismatch between the fetal head and the mother's pelvic brim [31,32].

**Mal-presentation**: Refers to any presentation other than vertex(brow, face, breech, transverse) [31,32].

**High birth weight**: Refers to the weight of baby at birth is 4000gm and above [31,32].

**Normal birth weight**: When the weight of baby at birth is between 2500gm-3999gm [31,32].

**Low birth weight**: Refers to the weight of baby at birth is between 1500gm-2499gm [31,32].

**Antenatal care follow-up**: Refers to the mother reported that she had visited any health institution during her recent pregnancy [33,34].

## Data collection tools and procedures

Face-to-face interviews were utilized to gather data, and a structured questionnaire adapted from other studies conducted in the Mettu Karl Referral Hospital, Harergehe, Halaba, Welega, and Adama [16,21,27,35,36] was used for data collection. In addition, the tool was validated in the previous studies. The English version of the questionnaire was translated to the local language (Afan Oromo) and then retranslated back to English by two language experts to ensure its consistency [S1 and S2 Files]. From November 10 to December 30, 2019, two BSc midwives (who work outside of the study area) and a supervisor collected data. After taking consent, the data collectors interviewed study participants and reviewed their respective clinical information from medical records.

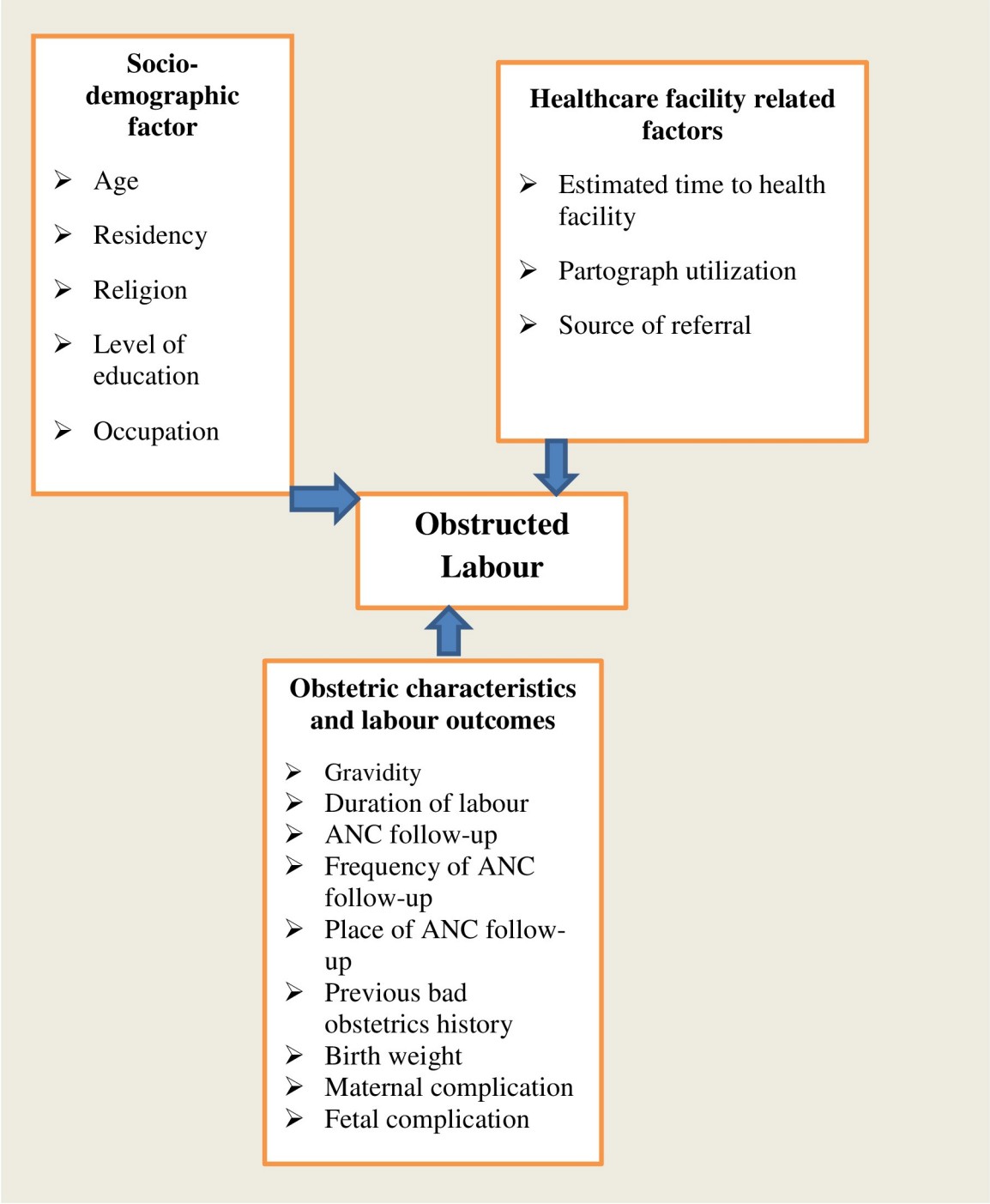

**Fig 1. A conceptual framework describing the outcome and explanatory variables of obstructed labour and its associated factors among women who gave birth at public health facilities in Mojo Town, Central Ethiopia, 2019.**

### Data quality control

The training was given to data collectors and the supervisor two days prior to data collection. The questionnaire was translated to the local language to make it clear to the participants. The

data collection process was thoroughly monitored by the supervisor on a daily basis. Moreover, data were checked for completeness, adequacy, and consistency before analysis.

## Data processing and analysis

The collected data were checked, coded, and entered into EpiData version 3.1 and then exported to SPSS version 23 for analysis. Descriptive statistics such as frequency, mean, and standard deviation were used to describe socio-demographic, obstetric, and healthcare characteristics. A binary logistic regression model was used to identify the association between the independent and the outcome variable. To verify the significant association, variables with a P-value < 0.25 in the bivariate model were re-entered into a multivariable logistic regression model. Finally, variables with a P-value of < 0.05 were considered statistically significant. The variance inflation factor (VIF) and tolerance tastes were used to check the presence of multicollinearity among the covariates. Moreover, the Hosmer-Lemeshow goodness of fit model was used to assess whether the number of expected events from the logistic regression model reflects the number of observed events in the data.

## Ethical consideration

An ethical clearance letter was obtained from the Institutional Ethical Review Board (IERB) of Adama Hospital Medical College. In addition, a permission letter was obtained from the Mojo Town health office and Mojo hospital prior to data collection. After a detailed explanation of the study's benefits and risks, verbal consent was obtained from each participant to assert willingness.

## Results

### Socio-demographic characteristics of participants

In this study, 318 subjects participated, with a response rate of 100%. The mean age of the participants was 25.6 with an SD ± 5.86. More than one-third (36.5%) of participants were aged 20–24 years. Two hundred four (63.9%) of the participants were urban residents. One hundred eighteen (37%) of them attended primary education. In terms of occupation, 189 (59.2%) participants were housewives [Table 1].

### Obstetric characteristics and labour outcomes

Nearly one-half (47.4%) of participants were primigravida. More than one-half of participants stayed less than twelve hours in labour. The majority, 303 (95.2%) of the participants had ANC follow-ups, of whom 244 (76.7%) had more than two-time follow-ups. One-third, 113 (35.5%) of participants had a previous bad obstetric history. Nearly one-quarter of participants experienced maternal and fetal complications [Table 2].

### Healthcare facility-related characteristics of participants

More than one-half of cases were completely followed-up with partograph by midwives. A total of 188 participants (59.1%) were referred from nearby health facilities. Moreover, more than three-fourths of participants lived within less than an hour's distance of a health facility [Table 3].

**Table 1. Socio-demographic characteristics of women who gave birth at public health facilities in Mojo town, Central Ethiopia, 2019 (n = 318).**

| Variables | Categories | Frequencies | Percentages (%) |
|---|---|---|---|
| Age (in year) | ≤ 19 | 41 | 12.9 |
| | 20–24 | 116 | 36.5 |
| | 25–29 | 98 | 30.8 |
| | 30–34 | 39 | 12.3 |
| | ≥ 35 | 24 | 7.5 |
| Residency | Urban | 204 | 64.1 |
| | Rural | 114 | 35.8 |
| Educational status | Uneducated | 54 | 17 |
| | Primary | 118 | 37.1 |
| | Secondary | 72 | 22.6 |
| | Collage and above | 74 | 23.3 |
| Religion | Orthodox | 209 | 65.5 |
| | Muslim | 36 | 11.3 |
| | Protestant | 46 | 14.4 |
| | Waqefata | 15 | 8.8 |
| Occupation | Private Organization | 32 | 10.0 |
| | Government employee/ employer | 26 | 8.2 |
| | Merchant | 28 | 8.8 |
| | Private own work | 30 | 9.4 |
| | House wife | 189 | 59.2 |
| | Others* | 13 | 4.1 |

Note:

* Daily labor workers and students.

## Prevalence and causes of obstructed labour

In this study, the prevalence of OL was 51 (16%) (95%CI: 14.25, 17.65). Of these, CPD 33 (66%), mal-presentation 11 (22%), and mal-position 7 (12%) were reported by the clinicians as the causes of OL [Fig 2].

## Factors associated with obstructed labour

A binary logistic regression analysis was done to identify factors associated with obstructed labour. In the bivariate analysis, variables such as age, residency, previous bad obstetric history, gravidity, duration of labour, and frequency of ANC follow-up were shown to be associated at a p-value of less than 0.25. In multivariate analysis (after controlling for potential confounders), variables such as gravidity, frequency of ANC follow-up, and duration of labour showed an independent association with OL.

Accordingly, primigravidae mothers were 7 times more likely to encounter obstructed labour when compared to multigravidae (AOR; 7.748, 95%CI: 2.128, 18.29). The odds of developing OL were 9.5 times higher in mothers who had one ANC follow-up compared to mothers who had more than two ANC follow-ups (AOR; 9.5, 95%CI: 1.91, 33.07).

Moreover, in this study, the likelihood of developing OL was 20% lower among mothers who stayed 12–24 hours in labour when compared to those who stayed for less than 12 hours (AOR: 0.20, 95%CI: 0.17, 0.87) [Table 4].

**Table 2. Obstetric characteristics and labour outcomes of women who gave birth at public health facilities in Mojo town, Central Ethiopia, 2019 (n = 318).**

| Variables | Categories | Frequencies | Percentages |
|---|---|---|---|
| **Gravidity** | Multigravidea | 167 | 52.6 |
| | Primigarvidea | 151 | 47.4 |
| **Duration of labour (in hour)** | <12 | 167 | 52.5 |
| | 12–24 | 115 | 36.1 |
| | >24 | 36 | 11.3 |
| **ANC follow-up** | Yes | 293 | 92.1 |
| | No | 25 | 4.7 |
| **Frequency of ANC follow-up (n = 293)** | >Two-times | 244 | 76.7 |
| | Two-times | 24 | 7.5 |
| | One-time | 25 | 7.9 |
| **Place of ANC follow-up** | Mojo hospital | 92 | 31.4 |
| | Other facilities | 201 | 68.6 |
| **Previous bad obstetrics history** | No | 205 | 64.5 |
| | Yes | 113 | 35.5 |
| **Apgar score** | 7–10[a] | 229 | 72 |
| | 4–6[b] | 69 | 21.7 |
| | 0–3[c] | 20 | 6.3 |
| **Birth weight (in grams)** | <2500 | 33 | 10.7 |
| | ≥ 2500–4000 | 247 | 77.7 |
| | >4000 | 37 | 11.6 |
| **Maternal complication** | No | 245 | 77.0 |
| | Yes | 73 | 23.0 |
| **Fetal complication** | No | 240 | 75.5 |
| | Yes | 78 | 24.5 |

Note:

[a]Reassuring,

[b]moderately abnormal,

[c]low.

## Discussion

Obstructed labour has been one of the significant causes of obstetric complications such as maternal and prenatal mortality and morbidity, especially in developing countries including Ethiopia. This study aimed to assess the prevalence, causes, and factors associated with obstructed labour in Mojo Town, Central Ethiopia. Accordingly, the prevalence of OL in the

**Table 3. The healthcare facility-related characteristics of mothers who gave birth at public health facilities in Mojo town, Central Ethiopia, 2019 (n = 318).**

| Variables | Categories | Frequencies | Percentages |
|---|---|---|---|
| **Estimated time to health facility (in hour)** | <1 hr. | 247 | 77.7 |
| | 1–2 hr. | 56 | 17.6 |
| | ≥ 3 hr. | 15 | 4.7 |
| **Source of referral** | Referral from health facility | 188 | 59.1 |
| | Self-referral | 39 | 12.3 |
| **Partograph utilization** | Completely filled | 165 | 51.9 |
| | Not completed | 100 | 31.4 |
| | Not filled at all | 53 | 16.6 |

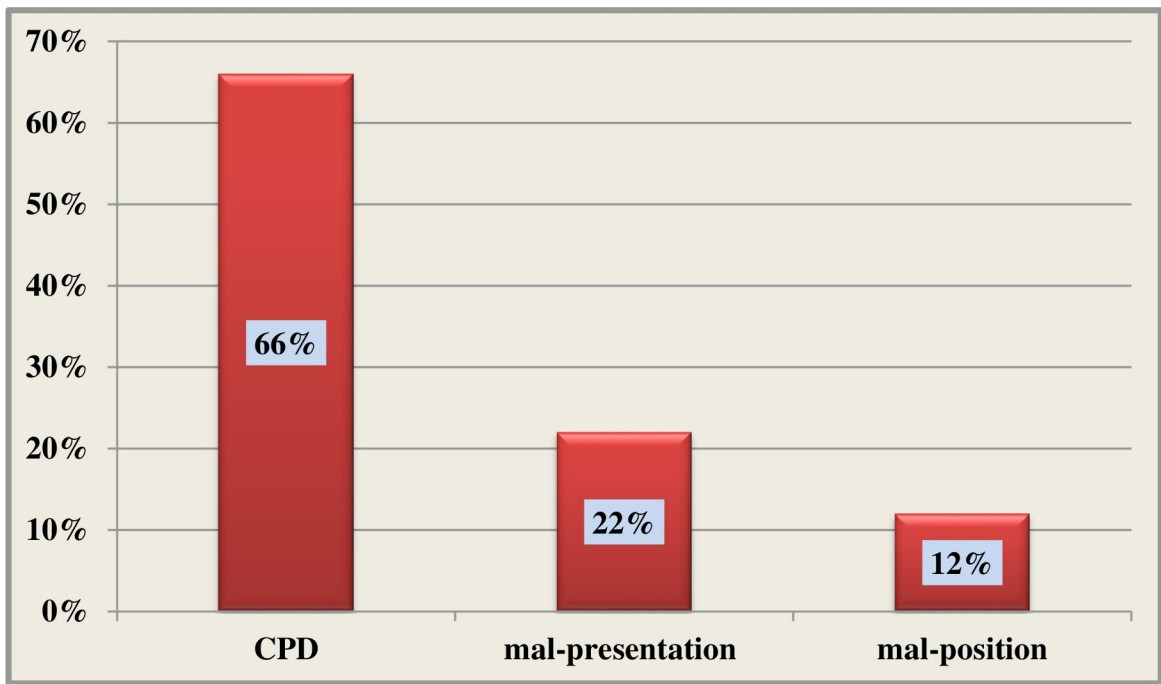

**Fig 2. Causes of obstructed labour among women gave birth at public health facilities in Mojo Town, Central Ethiopia, 2019.**

**Table 4. Factors associated with obstructed labour among women gave birth at public health facilities in Mojo town, Central Ethiopia, 2019 (n = 318).**

| Variables | Categories | Obstructed labour | | COR(95%CI) | AOR(95%CI) |
|---|---|---|---|---|---|
| | | Yes (%) | No (%) | | |
| **Age(in years)** | ≤ 19 | 21(6.6) | 20(6.3) | 0.88(0.33, 0.97) | 0.73(0.38, 0.91) |
| | 20–24 | 44(13.8) | 77(24.2) | 0.48(0.08, 0.79) | 0.62(0.15, 0.83) |
| | 25–29 | 38(11.9) | 55(17.3) | 0.58(0.12, 0.94) | 0.41(0.18, 0.97) |
| | 30–34 | 11(3.5) | 28(8.8) | 0.33(0.06, 0.67) | 0.31(0.03, 0.93) |
| | ≥ 35 | 13(4.1) | 11(3.5) | 1 | 1 |
| **Place of residence** | Urban | 77(24.2) | 127(40.0) | 1 | 1 |
| | Rural | 63(19.8) | 51(16.0) | 2.04(1.36, 5.85) | 1.65(0.99, 2.85) |
| **Previous bad obstetrics Hx** | No | 82(25.8) | 123(38.7) | 1 | 1 |
| | Yes | 61(19.2) | 52(16.3) | 1.76(1.01, 3.05) | 1.90(1.25, 3.63) |
| **Gravidity** | Primibravidae | 40(26.4) | 111(73.3) | 5.11(2.51,10.39) | 7.75 (2.12,18.29)** |
| | Multigravida | 11(6.6) | 156(93.4) | 1 | 1 |
| **Duration of Labor** | <12 hours | 6(3.5) | 161(96.4) | 1 | 1 |
| | 12–24 hours | 16(13.7) | 100(86.2) | 0.23(0.11, 0.92) | 0.20(0.17,0.87) * |
| | >24 hours | 29(80.5) | 7(19.4) | 0.01(0.009,1.10) | 0.03(0.02,1.05) |
| **Frequency of ANC Follow-up** | One | 16(64) | 9(36) | 7.58(2.32, 13.08.) | 9.50(1.91,33.07) * |
| | Two | 4(16) | 20(84) | 0.81(0.32, 1.15) | 0.76(0.27, 1.53) |
| | > two | 22(20.4) | 222(87) | 1 | 1 |

Note:

☞ *History,*

** *strongly significant association at p-value< 0.001,*

* *Significant association at p-value <0.05, and*

[1] *reference group.*

area was 16%. The current prevalence is lower compared to findings from the previous studies conducted in public hospitals in the Harergeh zone (34.30%), Halaba Kulito Hospital (18.6%), and West Wollega zone (18.1%) [21,27,35]. The possible reason for this discrepancy could be due to variations in the study design and period. It could also be tied to socio-demographic differences in the current and previous study populations.

The present prevalence is higher than in three studies conducted in India (a governmental medical college in Jhalawar (1.1%), Patna Medical College and Hospital (8.9%), and Hyderabad (3.61%)), Sokoto, Nigeria (2.0%), a community study from Uganda (10.5%), Adgrat zonal Hospitals (3.3%), Mizan-Aman General Hospital (7.95%), Mizan-Tepi University Teaching Hospital (15.6%), Mettu Karl Referral Hospital (4.1%), Jimma University Specialized Hospital (12.2%) and Adama Hospital Medical College (9.6%) [9,12,13,16,28,36–41]. The discrepancy could be related to the difference in the study design and socio-demographic variations among the current and previous study populations.

This study identified the causes of OL in our study area. Accordingly, CPD (66%), mal-presentation (22%), and mal-position (12%) were reported as causes of OL. This finding is consistent with the previous studies conducted in Bangladesh, India, Nigeria, Uganda, the Tigray region, Mizan Aman, Jimma, and Adama [12,13,22,28,36,38,40,42]. However, the current proportion is different from that of a study conducted in Bihar, India in which mal-position was the major cause of OL, followed by CPD, and mal-presentation [37]. The reason for this difference might be due to nutritional and socio-demographic variation in the two populations.

In this study, primigravidity was found to be an associated factor with OL. The odds of developing obstructed labour among primigravida were 7 times higher than compared to multigravida. This association was supported by studies conducted in Bangladesh, eastern Uganda, and Gimbi Town public hospitals [25,35,42]. This association might result from the psychological impact of the primiparous mother on the labour mechanism.

In the present study, the likelihood of encountering OL was 9.5 times higher among mothers who had one ANC follow-up compared to those who had more than two ANC follow-ups. This result was supported by a finding from a study conducted in Mizan Aman, Ethiopia [38]. This significance is tied to the scientific fact that women who had frequent ANC follow-ups could potentially benefit from early identification and prevention of OL prior to the onset of labour.

Moreover, the normal duration of labour was found to be protective for OL. Mothers who stayed 12–24 hours in labour were 20% less likely to develop OL. This finding is supported by a study conducted at Adama hospital [36]. The association resembles the truth that OL is pronounced in the prolonged duration of labour.

Despite the clinical and scientific plausibility, variables such as the weight of the baby and the age of the mother were not shown to be associated with OL in the current study. These variables are important attributes of OL, as evidenced by previous literature [11,36,42]. The possible reason for this difference might be due to a difference in the study designs of the current and previous studies.

Our study's strength comes from its full response rate. However, the cross-sectional nature of this study limits us from asserting a cause-effect relationship. Additionally, the reliability of each diagnosed case is questionable because this study relied solely on medical diagnoses to assess obstructed labor and its causes. We therefore suggest future researchers identify cause-and-effect relationships using rigorous designs, such as experimental studies.

## Conclusion

The prevalence of OL in the study area was higher than the majority of previous similar local and global studies. CPD, mal-presentation, and mal-position were reported as causes of OL. In

addition, factors such as gravidity, frequency of ANC follow-up, and duration of labour were significantly associated with OL. Therefore, the concerned entities need to work to strengthen early risk identification and counsel mothers on the importance of subsequent ANC-follows in the prevention of OL. Strategies including early detection of OL and management training for healthcare providers need to be emphasized in healthcare facilities. Additionally, we recommend researchers dig out other possible causes and risk factors of OL by using strong study designs.

## Supporting information

**S1 File. English version questionnaire.**
(DOCX)

**S2 File. Afan Oromo version questionnaire.**
(DOCX)

**S3 File. STROBE statement checklist.**
(DOCX)

**S1 Data.**
(XLS)

## Acknowledgments

The authors express our heartfelt gratitude to study participants for their willingness to take part in and squander their precious time in this study. Moreover, we would like to thank data collectors and supervisors for their devotion to collecting high-quality data.

## Author Contributions

**Conceptualization:** Tarekegn Girma, Wubishet Gezimu, Ababo Demeke.

**Data curation:** Tarekegn Girma, Wubishet Gezimu, Ababo Demeke.

**Formal analysis:** Tarekegn Girma, Wubishet Gezimu, Ababo Demeke.

**Investigation:** Tarekegn Girma, Wubishet Gezimu, Ababo Demeke.

**Methodology:** Tarekegn Girma, Wubishet Gezimu, Ababo Demeke.

**Software:** Tarekegn Girma, Wubishet Gezimu, Ababo Demeke.

**Supervision:** Tarekegn Girma, Wubishet Gezimu, Ababo Demeke.

**Validation:** Tarekegn Girma, Wubishet Gezimu, Ababo Demeke.

**Visualization:** Tarekegn Girma.

**Writing – original draft:** Tarekegn Girma, Wubishet Gezimu.

**Writing – review & editing:** Tarekegn Girma, Wubishet Gezimu, Ababo Demeke.

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
