## [Decision Letter · Decision Letter 0]

27 Jun 2022

PONE-D-21-26867Magnitude, Causes and Factors associated with Obstructed Labour among Mothers who Gave Birth at Public Health Facilities in Mojo Town, Mojo, Central Ethiopia, 2019PLOS ONE

Dear Dr. Gezimu,

Thank you for submitting your manuscript to PLOS ONE. After careful consideration, we feel that it has merit but does not fully meet PLOS ONE’s publication criteria as it currently stands. Therefore, we invite you to submit a revised version of the manuscript that addresses the points raised during the review process.

Specifically, the reviewers have multiple concerns including English language and missing details on methodology. Please have those concerns addressed point-by-point.

We look forward to receiving your revised manuscript.

Kind regards,

Jianhong Zhou

Staff Editor

PLOS ONE

Journal Requirements:

4. Please ensure that you refer to Figure 1 in your text as, if accepted, production will need this reference to link the reader to the figure.

Reviewers' comments:

Reviewer's Responses to Questions

**Comments to the Author**

1. Is the manuscript technically sound, and do the data support the conclusions?

Reviewer #1: Yes

Reviewer #2: Partly

Reviewer #3: No

2. Has the statistical analysis been performed appropriately and rigorously? 

Reviewer #1: Yes

Reviewer #2: Yes

Reviewer #3: No

3. Have the authors made all data underlying the findings in their manuscript fully available?

Reviewer #1: Yes

Reviewer #2: Yes

Reviewer #3: No

4. Is the manuscript presented in an intelligible fashion and written in standard English?

Reviewer #1: Yes

Reviewer #2: Yes

Reviewer #3: No

5. Review Comments to the Author

Reviewer #1: Review Comments to the Author

1. Needs English language correction.

2. It is a good effort and may help in reduction of such complications

Reviewer #2: 1) The area studied is of great clinical importance. Unfortunately, many women still experience labour complications and these contribute to the maternal morbidity and mortality.

2) The manuscript is relevant, it has numerous grammatical and typographical errors that need to be corrected.

3) The background/introduction has many sentences constituting paragraphs. Please coalesce most of them into relevant paragraphs and stratify them as follows:

a) Important definitions such as Obstructed labour, including the burden of obstructed labour globally, in Africa/sub-Saharan Africa and in Ethiopia

b) The causes of obstructed labour

c) The factors associated with obstructed labour

N.B: Not sure that the last sentence in the background about this being a pioneer study is relevant here. Which databases did the authors search to come to this conclusions?

4) In the methods, the use of the term cephalo-pelvic disproportion is misleading. What exactly caused the disproportion is the real cause of obstructed labour. Please, if possible, unpack the cephalo-pelvic disproportion variable.

5) Under birth weight, you write that a high birth weight is when the weight of baby at birth is 400gm and above. This is wrong. 400gm is too low to be a high fetal weight. I guess you meant 4000gm! Please correct this anomaly!

The normal birth weight range is also not 2500gm-399gm but 2500-3999gm! Also correct this anomaly!

6) In the analysis part, please state how categorical and continuous variables were analysed instead of clamping everything together.

7) In results, in Table 1 you state under occupation that the category "other" meant daily labour worker. Were there any other considerations under that category? If not, then just write daily labour worker directly in the table.

Under APGAR SCORE, what was the rationale of using the ranges the author uses? If possible, please stratify your APGAR SCORE as follows:

7–10: Reassuring

4–6: Moderately abnormal

0–3: Low

8) Under discussion, the content is there but needs to be improved. Please write your discussion as follows:

State your results, then results from comparable studies and the possible explanation for the similarities or differences between your study results and the quoted studies.

Otherwise your study is of great importance.

Reviewer #3: Review summary

Thank you very much for giving me the opportunity to review this paper. I agree that obstructed labour is an important clinical and public health problem in low resource settings. So, I don’t see what new information this manuscript is adding to scholarship on obstructed labour. The methods described by the authors are not sound, and the manuscript is not well-written. I have a couple of issues regarding this work.

Major issues

1. Use of the word magnitude in the title is not specific enough, because this is very difficult to measure. The authors should consider revising this to prevalence.

2. The introduction is very shallow and not informative because the authors have not reviewed and distilled the available literature on obstructed labour, yet it is a well-studied subject.

3. Subsequently, the gap/problem as well as the justification is not clearly defined in the introduction.

4. The stated objective is not SMART.

5. Generally, the methods are poorly reported, the authors should make use of the STOBE checklist to improve on the transparency of reporting. It will also help to make it more comprehensive.

6. Line 131- 132 is not clear, please check it and revise the grammar and sentence structure. Please do this for the rest of the document as well.

7. The primary outcomes are two, but they are not well defined. It is important to report how, who and when the diagnosis of labour was made? Are there any guidelines for diagnosis of obstructed labour in your facility?

8. You did not include a section on study procedures. So, it is not clear when, how and who collected the data. When was the consent obtained? Did you review patient files or interview participants for data collection?

9. The results are not well presented. For instance, in Table 2, why do you have three variables on the same factor of ANC? What are they showing? Generally, there is a big problem with the categorizations on several variables; it is so arbitrary and unconventional. In table 3, under partograph is another example of the same.

10. Table 4 is presenting outcomes of obstructed labour; this is not one of the objectives for this study. This is completely new and not acceptable.

11. In the discussion, it is hard to know what the key message is. This has also affected the conclusions and recommendations that have been advanced by the authors.

Minor issues

1. Kindly pay attention to the grammatical errors and typos that are all over the document

2. Have a look at the references as well and ensure that they are more accurate.

3. Many sentences are either incomplete or not meaningful.

4. The paragraphs are not well structured, it is not proper to have one sentence standing alone as a paragraph.

6. PLOS authors have the option to publish the peer review history of their article (what does this mean?). If published, this will include your full peer review and any attached files.

Reviewer #1: No

Reviewer #2: **Yes: **Dr. Joseph Ngonzi

Reviewer #3: **Yes: **Milton Musaba

---

## [Author Response · Author response to Decision Letter 0]

27 Jul 2022

Response to Reviewers 

Title: “Magnitude, Causes, and Factors associated with Obstructed Labour among Mothers who Gave Birth at Public Health Facilities in Mojo Town, Mojo, Central Ethiopia, 2019” 

Manuscript ID: PONE-D-21-26867

Dear, Editor and reviewers, we (authors) would like to thank you for your time and fruitful suggestions. We respond and corrected all raised comments. Hereunder we enclosed all the answers and corrections point by point.

Response to Editor’s comments

Authors’ response: Dear Editor, Thanks a lot for your guidance with PLOS ONE's guidelines! We have corrected the entire manuscript as per PLOS ONE's requirements.

2. In your Data Availability statement, you have not specified where the minimal data set underlying the results described in your manuscript can be found.

Authors’ response: Great! We have updated the data availability statement and included all the data used in the study as supporting information in the revised manuscript.

Authors’ responses: Dear, Thank you so much for your suggestion! We made changes to the data availability statement and described it in the cover latter. 

4. Please ensure that you refer to Figure 1 in your text as, if accepted, production will need this reference to link the reader to the figure 

Authors’ response: Dear Editor, Thanks for reminding us of our mistake. We have referred to figure 1 under the study variable sub-section of methodology.

5. Please include captions for your Supporting Information files at the end of your manuscript, and update any in-text citations to match accordingly. 

Authors’ response: Thanks for reminding us! We have included supporting information and cited its corresponding captions in the text. 

Response to Reviewer #1 comments

1. Needs English language correction.

Authors’ response: Dear reviewer, we are grateful for your suggestion. We have corrected all the language errors in our manuscript as highlighted in the track change file.

2. It is a good effort and may help in reduction of such complications

Authors’ response: Thanks a lot for your constructive comments!

Response to reviewer #2 comments

1. The area studied is of great clinical importance. Unfortunately, many women still experience labour complications and these contribute to the maternal morbidity and mortality

Authors’ response: Dear Reviewer, Thank you so much for acknowledging our study outcome (obstructed labour), one of the causes of maternal mortality and morbidity. 

2. The manuscript is relevant; it has numerous grammatical and typographical errors that need to be corrected.

Authors’ response: Thank you for your fruitful comments! 

We have corrected all the typographic and grammatical errors

3. The background/introduction has many sentences constituting paragraphs. Please coalesce most of them into relevant paragraphs and stratify them as follows:

a) Important definitions such as Obstructed labour, including the burden of obstructed labour globally, in Africa/sub-Saharan Africa and in Ethiopia

b) The causes of obstructed labour

c) The factors associated with obstructed labour

N.B: Not sure that the last sentence in the background about this being a pioneer study is relevant here. Which databases did the authors search to come to this conclusions?

Authors’ response: Great! We made significant changes to the introductory parts of our paper as per your suggestions. We stratified it as: definition and burden of OL, causes of OL factors associated, strategies that have been taken to prevent OL, and gaps in data about OL in the area. ‘This is a pioneer study which assessed causes of obstructed 91 labour in the Central Ethiopia’ just to express the limitation of evidence specifically, in our study setting. We have removed this section.

4. In the methods, the use of the term cephalo-pelvic disproportion is misleading. What exactly caused the disproportion is the real cause of obstructed labour. Please, if possible, unpack the cephalo-pelvic disproportion variable. 

Authors’ response: OH GREAT! Really, it is insightful suggestion. Thanks! According to different literatures and guidelines CPD is taken as the first and leading cause of obstructed labour (as cited in the background section). In fact, there are root causes of CPD which are described as risk factor for obstructed labour such as maternal age, fetal macrosomia (hydrocephaly), contracted pelvis, etc.

5. Under birth weight, you write that a high birth weight is when the weight of baby at birth is 400gm and above. This is wrong. 400gm is too low to be a high fetal weight. I guess you meant 4000gm! Please correct this anomaly! The normal birth weight range is also not 2500gm-399gm but 2500-3999gm! Also correct this anomaly! 

Authors’ responses: Dear, we are sorry for the mistake we made! It was a typing error! We meant 4000gm and 3999. We have corrected it. Thanks a lot for reminding us!

6. In the analysis part, please state how categorical and continuous variables were analysed instead of clamping everything together.

Authors’ responses: Excellent! Dear, the majority of the independent variables handled in our study were categorical in form, and before analysis, we also categorized continuous variables such age, labour time, and estimated distance to healthcare facilities.

7. In results, in Table 1 you state under occupation that the category "other" meant daily labour worker. Were there any other considerations under that category? If not, then just write daily labour worker directly in the table.

Under APGAR SCORE, what was the rationale of using the ranges the author uses? If possible, please stratify your APGAR SCORE as follows:

7–10: Reassuring

4–6: Moderately abnormal

0–3: Low

Authors’ responses: I really appreciate your helpful suggestion. Of course, there is another factor under the other category of Table 1 that was overlooked when typing, namely "students."

Dear, we have also changed the APGAR SCORE category in accordance with your recommendation.

8. Under discussion, the content is there but needs to be improved. Please write your discussion as follows: State your results, then results from comparable studies and the possible explanation for the similarities or differences between your study results and the quoted studies.

Authors’ responses: Dear reviewer, thanks for your guidance! We wrote the discussion section as per your recommendations!

Response to reviewer #3 comments

Major issues 

1. Use of the word magnitude in the title is not specific enough, because this is very difficult to measure. The authors should consider revising this to prevalence 

 Authors’ responses: Dear Reviewer, We value your insightful suggestion. We have replaced the word "magnitude" with "prevalence" in the title section.

2. The introduction is very shallow and not informative because the authors have not reviewed and distilled the available literature on obstructed labour, yet it is a well-studied subject

Authors’ responses: Great! We made thorough revision to the introduction part and incorporated all relevant data to our study’s outcome interest. Thanks in advance!

3. Subsequently, the gap/problem as well as the justification is not clearly defined in the introduction.

Authors’ responses: Dear review, thank you for your concern over this important section! We have updated this section and we hope you will get the gaps and justification we articulated. Thanks!

4. The stated objective is not SMART.

Authors’ responses: Dear, thanks once more! We have corrected this section. We have corrected the study objectives and made them SMART.

5. Generally, the methods are poorly reported, the authors should make use of the STOBE checklist to improve on the transparency of reporting. It will also help to make it more comprehensive. 

Authors’ responses: Dear, thank you in advance for your suggestions! We have filled and uploaded the STROBE checklist with supportive information. 

6. Line 131- 132 is not clear, please check it and revise the grammar and sentence structure. Please do this for the rest of the document as well.

Authors’ responses: Dear, Thanks a lot! We have corrected the grammar and sentence structures of the mentioned section and in all the rest of the document.

7. The primary outcomes are two, but they are not well defined. It is important to report how, who and when the diagnosis of labour was made? Are there any guidelines for diagnosis of obstructed labour in your facility? 

Authors’ responses: Dear reviewer, Thank you so much for your perceptive comments! The two outcomes of this study, Prevalence of obstructed labour and causes of obstructed labour were diagnosed by physicians. In Ethiopia, physicians use the Federal guideline for diagnosis and management of obstetrics cases.

(Federal Democratic Republic of Ethiopia Ministry of Health: Management Protocol on Selected obstetrics topics, January, 2010).

8. You did not include a section on study procedures. So, it is not clear when, how and who collected the data. When was the consent obtained? Did you review patient files or interview participants for data collection?

Authors’ responses: Great! Dear reviewer, we have described the study procedures in the ‘data collection tools and procedure’ sub-section of methods and material. 

The data were collected by two BSc Midwives (who works out of the study area) and a supervisor from 10 November to 30 December 2019. 

Consent was taken from each participant before interview.

Great! We have used both interview and patient file review. 

After data collectors taken consent, they interviewed study participants and reviewed the respective clinical information from medical record.

9. The results are not well presented. For instance, in Table 2, why do you have three variables on the same factor of ANC? What are they showing? Generally, there is a big problem with the categorizations on several variables; it is so arbitrary and unconventional. In table 3, under partograph is another example of the same. 

 Authors’ responses: Dear reviewer, Thank you so much for your insightful comments! Dear, we have unpacked the category of having ANC information and left having ANC follow-up and frequency of ANC follow-up. Because, evidence shows that frequency of ANC has association with obstructed labour. Mostly, those mothers who had less ANC follow-up were risky for OL. We have corrected all of the commented sections.

10. Table 4 is presenting outcomes of obstructed labour; this is not one of the objectives for this study. This is completely new and not acceptable. 

Authors’ responses: Dear, we appreciate you for your intellectual suggestion! 

Of course, outcome of obstructed labour was not our study objective but we included the feto-maternal outcome of participants in the study area as an explanatory variable. Because, as literatures shown, feto-maternal outcomes such as birth weight affects labour outcomes. Dear, to make it more clear and attractive for readers, we have merged the important feto-maternal outcome variables to the obstetrics characterises section of the findings. 

11. In the discussion, it is hard to know what the key message is. This has also affected the conclusions and recommendations that have been advanced by the authors. Authors’ responses: Dear, thanks for your helpful suggestion! We revised the discussion section. We concluded what we found and forwarded as per our findings. Thanks in advance again!

Minor issues 

1. Kindly pay attention to the grammatical errors and typos that are all over the document Authors’ responses: Dear reviewer, thanks for your constructive comments! We have corrected all the grammatical errors and typos throughout the document.

2. Have a look at the references as well and ensure that they are more accurate. 

Authors’ responses: Thank you! We have checked and corrected it.

3. Many sentences are either incomplete or not meaningful. 

Authors’ responses: Thank you dear! We have checked and corrected all the sentence errors all over the manuscript.

4. The paragraphs are not well structured, it is not proper to have one sentence standing alone as a paragraph. 

Authors’ responses: We greatly appreciate you! All of the paragraphs have been rearranged, and we have fixed all the mentioned errors.

---

## [Decision Letter · Decision Letter 1]

8 Sep 2022

PONE-D-21-26867R1Prevalence, causes, and factors associated with obstructed labour among mothers who gave birth at public health facilities in Mojo Town, Central Ethiopia, 2019PLOS ONE

Dear Authors,

Thank you for submitting your manuscript to PLOS ONE. After careful consideration, we feel that it has merit but does not fully meet PLOS ONE’s publication criteria as it currently stands. Therefore, we invite you to submit a revised version of the manuscript that addresses the points raised during the review process.

ACADEMIC EDITOR: The introduction part sounds clear and provides the study rationale well. However, I suggest rewriting the last paragraph (lines 95-101 in the revised document). The issue is that you write "the aim of this study was" (singular), however below you stated THREE aims (plural). So, this should sound more clear.  Moreover, the writing style should be improved in this paragraph to be closer to the academic writing style. It is better to remove the numbering (1,2,3) and present it in plain text.In the Methods part, you described "Operational definition" (singular), but in the text there are many definitions, so the subheading should be "Operational definitions".Please make sure you've followed the PLOS ONE journal's requirements https://journals.plos.org/plosone/s/submission-guidelines ==============================

We look forward to receiving your revised manuscript.

Kind regards,

Gulzhanat Aimagambetova

Academic Editor

PLOS ONE

Journal Requirements:

Reviewers' comments:

Reviewer's Responses to Questions

**Comments to the Author**

1. If the authors have adequately addressed your comments raised in a previous round of review and you feel that this manuscript is now acceptable for publication, you may indicate that here to bypass the “Comments to the Author” section, enter your conflict of interest statement in the “Confidential to Editor” section, and submit your "Accept" recommendation.

Reviewer #1: (No Response)

Reviewer #2: All comments have been addressed

2. Is the manuscript technically sound, and do the data support the conclusions?

Reviewer #1: Yes

Reviewer #2: Yes

3. Has the statistical analysis been performed appropriately and rigorously? 

Reviewer #1: Yes

Reviewer #2: Yes

4. Have the authors made all data underlying the findings in their manuscript fully available?

Reviewer #1: Yes

Reviewer #2: Yes

5. Is the manuscript presented in an intelligible fashion and written in standard English?

Reviewer #1: No

Reviewer #2: Yes

6. Review Comments to the Author

Reviewer #1: Overall a good effort in writing the manuscript, however the comparison with other studies should have been presented in a standard way. Recommendation must mention the strategies to prevent obstructed labour in hospital settings such as training of birth attendants to monitor and identify cases who may land up in obstructed labour.

Reviewer #2: The author has addressed the comments raised by the reviewers. The manuscript reads much better than before.

7. PLOS authors have the option to publish the peer review history of their article (what does this mean?). If published, this will include your full peer review and any attached files.

Reviewer #1: No

Reviewer #2: **Yes: **Dr. Joseph Ngonzi

---

## [Author Response · Author response to Decision Letter 1]

10 Sep 2022

Response to Reviewers 

PONE-D-21-26867R1

Prevalence, causes, and factors associated with obstructed labour among mothers who gave birth at public health facilities in Mojo Town, Central Ethiopia, 2019: A cross-sectional study

Dear Dr. Gulzhanat Aimagambetova (Editor) and reviewers, we appreciate your valuable time with our article. Thanks for your intellectual comments and suggestions. We have revised our article and responded point-by-point to all the editor and reviewers comments and suggestions hereunder.

ACADEMIC EDITOR: 

The introduction part sounds clear and provides the study rationale well. However, I suggest rewriting the last paragraph (lines 95-101 in the revised document). The issue is that you write "the aim of this study was" (singular), however below you stated THREE aims (plural). So, this should sound more clear. Moreover, the writing style should be improved in this paragraph to be closer to the academic writing style. It is better to remove the numbering (1,2,3) and present it in plain text.

Authors’ response: Dear Editor, Thank you so much for your intelligent suggestion. We have changed ‘aim’ to ‘aims’ and ‘was’ to ‘were’ in the mentioned section. Also, we have removed the numbering from the objectives of the study and presented it in plain text (highlighted on page no. 5; line no. 92-94). 

In the Methods part, you described "Operational definition" (singular), but in the text there are many definitions, so the subheading should be "Operational definitions".

Authors’ response: Dear Editor, Thanks once again. We have changed "Operational definition" to be "Operational definitions"(highlighted on page no. 7 and line no. 141). 

Please make sure you've followed the PLOS ONE journal's requirements 

Authors’ response: Dear Editor, we have followed the PLOS ONE journal's requirements. 

Review Comments to the Author

Reviewer #1: Overall a good effort in writing the manuscript, however the comparison with other studies should have been presented in a standard way. 

Authors’ response: Dear reviewer, thank you so much for your concern on this important section. We have corrected the mentioned section (highlighted on Page number 17; line number 290 and 291).

Recommendation must mention the strategies to prevent obstructed labour in hospital settings such as training of birth attendants to monitor and identify cases who may land up in obstructed labour.

Authors’ response: Dear reviewer, thank you so much for your valuable suggestions. We have mentioned the strategies to prevent obstructed labour in hospital settings as per your suggestion (highlighted on page number 17; line number 295 and 296). 

Reviewer #2: The author has addressed the comments raised by the reviewers. The manuscript reads much better than before.

Authors’ response: Dear reviewer, thank you for your time with article. 

Journal Requirements:

Authors’ response: Dear, we have revised the reference lists and completed and corrected some of missed contents of the references such as sure name, page number, DOI, Journals, and publishers. Additionally, we have replaced Reference no. 41 by a new relevant reference. The previous reference was incorrect and unintentionally placed during automated citing by Mendeley library.

Thanks all for your time!

---

## [Editor Report · Decision Letter 2]

13 Sep 2022

Prevalence, causes, and factors associated with obstructed labour among mothers who gave birth at public health facilities in Mojo Town, Central Ethiopia, 2019: A cross-sectional study

PONE-D-21-26867R2

Dear Authors,

We’re pleased to inform you that your manuscript has been judged scientifically suitable for publication and will be formally accepted for publication once it meets all outstanding technical requirements.

Kind regards,

Gulzhanat Aimagambetova

Academic Editor

PLOS ONE

---

## [Editor Report · Acceptance letter]

14 Sep 2022

PONE-D-21-26867R2 

Prevalence, causes, and factors associated with obstructed labour among mothers who gave birth at public health facilities in Mojo Town, Central Ethiopia, 2019: A cross-sectional study 

Dear Dr. Gezimu:

I'm pleased to inform you that your manuscript has been deemed suitable for publication in PLOS ONE. Congratulations! Your manuscript is now with our production department. 

Kind regards, 

on behalf of

Dr. Gulzhanat Aimagambetova 

Academic Editor

PLOS ONE